# Degradation of Tetracycline with Photocatalysis by CeO_2_-Loaded Soybean Powder Carbon

**DOI:** 10.3390/nano13061076

**Published:** 2023-03-16

**Authors:** Xinze He, Wenzhen Qin, Yu Xie

**Affiliations:** 1College of Environment and Chemical Engineering, Nanchang Hangkong University, Nanchang 330063, China; 2School of Materials Science and Engineering, Nanchang Hangkong University, Nanchang 330063, China

**Keywords:** photocatalytic, cerium dioxide, soybean powder carbon, organic pollutants

## Abstract

In the process of using photocatalysts to treat tetracycline (TC) wastewater, the degradation efficiency of soybean powder carbon material (SPC) can be improved by loading it with cerium oxide (CeO_2_). In this study, firstly, SPC was modified by phytic acid. Then, the CeO_2_ was deposited on modified SPC using the self-assembly method. Catalyzed cerium (III) nitrate hexahydrate (CeH_3_NO_4_) was treated with alkali and calcined at 600 °C under nitrogen. XRD, XPS, SEM, EDS, UV-VIS /DRS, FTIR, PL and N_2_ adsorption–desorption methods were used to characterize the crystal structure, chemical composition, morphology, surface physical and chemical properties. The effects of catalyst dosage, monomer contrast, pH value and co-existing anions on TC oxidation degradation were investigated, and the reaction mechanism of a 600 Ce-SPC photocatalytic reaction system was discussed. The results show that the 600 Ce-SPC composite presents uneven gully morphology, which is similar to the natural “briquettes”. The degradation efficiency of 600 Ce-SPC reached about 99% at 60 min under light irradiation when the optimal catalyst dosage and pH were 20 mg and 7. Meanwhile, the reusability of the 600 Ce-SPC samples showed good stability and catalytic activity after four cycles.

## 1. Introduction

In recent years, the use of antibiotics has reached its peak. Tetracycline (TC) is the most widely used and its consumption ranks second in the world [1]. This is because TC is not only used in clinical medicine in some countries, but it is also often used as a growth promoter [2]. However, due to overuse, TC cannot be effectively broken down and absorbed in living organisms, resulting in large amounts of TC being discharged into the aquatic environment [3]. Antibiotics can induce bacteria to develop resistance at low levels for long periods of time, seriously threatening ecosystems and human health [4]. For the treatment of wastewater containing TC, adsorption, biodegradation and electrochemical oxidation are the most common methods [5]. However, due to the stable aromatic ring and functional group molecular structure of TC, its removal efficiency is relatively low under natural conditions [6,7]. Non-degraded physical treatments (such as adsorption and filtration) can only separate TC from wastewater [8,9], while conventional chemical oxidation (such as chlorine oxidation and ozone oxidation) is prone to forming carcinogenic chlorination byproducts and has high operating costs [10]. In contrast, photocatalyst, which consists of photoexcitation and surface catalysis, is conducive to environmental protection and sustainable development [11].

Among variously reported photocatalysts, CeO_2_, a typical indirect transition N-type semiconductor, has a unique cerium electronic structure ([Xe]4fl5d16s2). CeO_2_ photocatalyst is favorable for charge transfer reactions because of its unique REDOX charge transfer between Ce^4+^ and Ce^3+^ oxidation states [12]. Pure CeO_2_ has a wide band gap of about 3.2 eV and mainly absorbs UV light. Due to the transformation of intrinsic charge from O 2p orbital to Ce 4f orbital, CeO_2_ has strong absorption at 350 nm, and its overall catalytic activity is closely related to the Ce^3+^ content converted to Ce^4+^ through lattice collapse during the calcination of CeO_2_ [13]. Due to insufficient catalytic active sites and low product selectivity in the catalysis of Ce, its catalytic effect is restricted, so biochar was used as the base material to improve the catalytic efficiency of CeO_2_. In recent years, CeO_2_ microspheres loaded on biological carbon have shown a higher surface and better visible light catalytic degradation efficiency of acid orange than pure cerium dioxide [14]. Duangdao Channei et al. [15] used CeO_2_/SCB composite catalysts to improve the photocatalytic degradation of dye. Alireza Khataee [16] reported the sonocatalytic and degradation performance of CeO_2_@biochar (CeO_2_-H@BC) nanocomposite, which provided us with the idea of proposing Ce loading in soybean flour.

Biochar has a high carbon content, abundant pore structure, large specific surface area and strong adsorption capacity, which are widely used in agriculture, ecological restoration and environmental protection. In addition, carboxyl phenol hydroxyl, acid anhydride and other groups on biochar surfaces can catalyze H_2_O_2_ to produce free OH·, which promotes the degradation of organic pollutants. However, traditional biochar still has some shortcomings, such as low adsorption capacity for pollutants, small amounts of pollutants (heavy metals) and difficulty separating from the environment. These factors limit the popularization and application of biochar. It is worth noting that the modification of biochar can effectively change its adsorption and degradation of pollutants. At present, there are many methods to modify biochar. Common physical modification methods include steam activation, ultraviolet radiation, ball milling, freeze–thaw cycle, etc. [17]. Chemical modification, such as acid modification [18], alkali modification [19] and organic reagent modification [20], is the most common method of biochar modification. For example, nitric acid modification can increase the carboxyl group and negative charge content of biochar and increase the adsorption capacity of U (VI) by 40 times [21]. Phytic acid, an additive with multiple functions, accelerates the formation of a hierarchical pore structure on the macro level. However, the application of phosphorus-modified soybean powder material is rarely mentioned.

In this study, the phytic acid modified soybean powder carbon was used as a carrier and closely combined with cerium dioxide under freeze-drying conditions. Although soybean powder carbon can promote environmental protection, it does not have good catalytic performance and can only be used as a carrier. The 600 Ce-SPC caused Ce nanoscale microspheres to disperse on the surface of soybean powder carbon and obtained a better catalytic performance and a good photoresponse. Techniques with broad application to characterize the physicochemical properties of CeO_2_/SPC include BET, SEM, FTIR, XRD and XPS. The objectives of this study were to investigate the efficiency and mechanism of the process of the oxidative degradation of CeO_2_/SPC composites, and to identify the feasible method for the degradation of TC.

## 2. Materials and Methods

### 2.1. Materials

Phytic acid (PA) was purchased from Shandong Sennuo, Jining, Shangdong, China. Ethanol was purchased from Sinopharm Chemical Reagent Co. Ltd., Shanghai, China. Sodium hydroxide and cerium (III) nitrate hexahydrate were purchased from Shanghai Macklin Biochemical Co. Ltd., Shanghai, China. Soybean powder carbon (SPC) was purchased from Lianfeng Agricultural Products Deep Processing, Liangyungang, Jiangsu, China. Deionized water was prepared in a laboratory.

### 2.2. Preparation of Photocatalysts

The soybean powder carbon was washed with deionized water and ethanol and dried in the oven at 60 °C. Then, the dried soybean powder carbon materials were placed into the porcelain boat at 200 °C for 30 min. Subsequently, the materials were heated to 600 °C for 150 min. After cooling and grinding, samples were obtained and named SPC.

PA-SPC was prepared by the following steps. First, in order to remove the impurities, the soybean powder carbon materials were washed with deionized water and ethanol several times, and then dried in the oven at 60 °C. Three grams of dried soybean carbon materials was added to 50 mL of deionized water and ultrasonic for 15 min to ensure they were evenly distributed. Then, 25 mL of phytic acid (PA) was added to the soybean carbon materials suspension and stirred at 700 rmp for 10 min. Next, in order to adjust the pH of the solution to 12, 0.2 M of NaOH solution was added and stirred at room temperature for 8 h. Subsequently, the samples were centrifuged and dried at 60 °C. Then, the dried samples were placed into the porcelain boat and heated at 200 °C for 30 min, then heated at 600 °C. After cooling and grinding, PA-SPC samples were obtained.

The Ce-SPC composites were prepared by the one-pot method. First, 3 g of soybean powder was placed in 50 mL of deionized water with ultrasonic for 15 min to ensure the soybean powder was evenly distributed. Then, 25 mL of phytic acid solution was added into the above suspension with ultrasonic for 10 min. Subsequently, NaOH solution (0.2 M) was added to the suspension to adjust the pH value of the suspension with stirring for 8 h, and the suspension was named solution A. Then, 2 g of Ce (NO_3_)_3_·6H_2_O was dissolved in 35 mL of deionized water, after which the Ce(NO_3_)_3_ solution was poured into solution A with magnetic stirring for 1 h. Then, the mixture was transferred to the refrigerator and solidified into ice cubes. Afterwards, the ice cube samples were placed in a freeze-dryer and dried for 48 h. Then, the freeze-dried materials were heated to 200 °C for 30 min at the rate of 5 °C/min under pure nitrogen. Finally, the different Ce-SPC composites with porous structure were obtained after calcining at 450, 500 and 600 °C for 2 h, separately, and the samples were named X Ce-SPC (X = 450, 500 and 600). 

### 2.3. Characterization of Photocatalysts

The material’s crystal structure was measured by X-ray diffraction (XRD, CuKα (λ = 1.5406 Å BRUKER, Ettlingen, Germany). The compositions of the Ce-SPC surface were analyzed through a Fourier transform infrared spectra (FTIR, VERTEX 70,BRUKER, Ettlingen, Germany). The X-ray photoelectron spectroscopy (XPS, Axis Ultra DLD) was utilized to study the surface chemical state and molecular structure of samples.The surface morphology and microstructure of the samples were analyzed by Scanning Electron Microscope (SEM, FEI Quanta 250, Hillsboro, OR, USA). The optical properties of samples were detected by the UV-Vis spectrophotometer (Spec-3700 DUV, Shimadzu, Kyoto, Japan).The photoluminescence spectroscopy (PL) of the material was determined by a fluorescence emitter (PL, Hitachi F4500, Chiyoda, Japan). Electrochemical impedance spectroscopy (EIS) and photocurrent testing of materials are performed using a traditional three-electrode electrochemical workstation (CHI 660D). The BET specific surface area of samples was recorded on a surface area and pore size distribution analyzer (Quantachrome NOVA 2000e, Boynton Beach, FL, USA).

### 2.4. Evaluation of Photocatalytic Activity

The photocatalytic activity of SPC, PA-SPC and X Ce-SPC was studied by degradation tetracycline (TC), which was conducted in conical flasks. Some of the catalysts were added to a 100 mL TC solution (10 mg/L), with stirring for 60 min under dark conditions, to reach adsorption equilibrium. The initial pH was adjusted to 6.9 ± 0.2 with NaOH. A lamp (300 W Xe) with a cutoff filter of 420 nm UV light was used as the visible light source. After the light reaction was turned on, the 3.0 mL reaction solution was removed at the same time interval. After centrifugation, the supernatant was taken and detected by the UV-vis spectrophotometer. The degradation efficiency (*η*) was simulated by the pseudo-first kinetic model, as follows:$$\eta =\frac{C_{0}-C}{C_{0}}*100\%$$
where *C*_0_ is the initial concentration of tetracycline; *C* is the concentration of residual tetracycline. The effect of the dosage of the catalysts (10, 15 and 20 mg), the concentration in TC 50 mg/L and the pH of the suspension (3, 5, 7, 9 and 11), were evaluated. In addition, the effects of Cl^−^ (10, 20, 40 mM), NO_3_^−^ (10, 20, 40 mM), HCO_3_^−^ (10, 20, 40 mM) and humic acid (1, 5, 10 mg/L) were investigated. The reactive oxygen species (ROS) generated in the catalytic degradation process of TC were measured via scavenging experiments. The scavengers of methanol (MA, 20 mM), tertiary butanol (TBA,10 mM), p-benzoquinone (BQ, 50 mM) and furfuryl alcohol (FFA, 20 mM) were used to quench ·OH/SO_2_^−^, ·OH, O_2_^−^ and 1O_2_, respectively. Furthermore, the reusability of the catalyst was employed to evaluate the stability of the 600 Ce-SPC catalyst. The 600 Ce-SPC catalyst was recycled by washing, centrifuging (8000 rpm for 4 min) and drying for the next experiment without any regeneration treatment. All the experiments were conducted in triplicate.

## 3. Results and Discussion

### 3.1. Characteristic of Catalysts

The crystal phases of SPC, PA-SPC, 600 Ce-SPC, 500 Ce-SPC and 450 Ce-SPC were investigated by XRD. As shown in Figure 1a, the peaks of the SPC samples at 24°, 31.84° and 34.52° correspond to the planes of (0), (100) and (002) [22], respectively. After phytic acid treatment, the peaks of the PA-SPC samples at 47.63° and 43° correspond to the planes of (102) and (101). Further, the intensities of are PA-SPC improved, implying that the crystalline material is somewhat more ordered [23,24]. As shown in Figure 1b, after depositing CeO_2_, all the peaks of the CeO_2_/SPC samples (600 Ce-SPC, 500 Ce-SPC and 450 Ce-SPC) display the typical cubic fluorite structure of CeO_2_ (JCPDS 34-0349) [25], with characteristic peaks at the 2θ values of 28.6°, 33.1°, 47.6°, 56.4°, 59.1°, 69.5°, 76.7° and 79.1° (Figure 1b). The results indicate that the CeO_2_ samples are successfully deposited on. It is noteworthy that the peak intensities of the samples increase with the temperature increase, indicating a high crystal structure at higher temperatures.

An FT-IR analysis was employed to study the chemical bonding of the materials, as displayed in Figure 1c. For the PA-SPC and Ce-SPC samples, there was an obvious vibration at approximately 977 cm^−1^, which indicated the existence of a C-P characteristic peak [26]. This is because phytic acid was used to modify SPC. In the FT-IR spectrum of SPC, the broad peaks at 1000–1100^−1^ [27] and 900–1000^−1^ [28] correspond with C-O and R-CH=CH, respectively [29]. However, in the comparison between SPC and PA-SPC, it can be observed that the peak of PA-SPC becomes sharper and more pronounced in the 1300–1140^−1^ range, where it represents the P=O group from the P element of phosphoric acid [30]. Moreover, PA-SPC has some more obvious peaks. The relevant characteristic absorbing peaks appear at 3015 cm^−1^ (Cp ring) [31] and 1509 cm^−1^ [32](aromatic compounds and Cp ring), which is attributed to the decomposition of Ce (NO_3_)_3_·6H_2_O-containing compounds at a high temperature. In addition, compared with the FT-IR spectra of SPC and PA-SPC, it is clear that the deposition of CeO_2_ on SPC gives rise to the emergence of new peaks at 1382 cm^−1^ [33] and 465 cm^−1^ [34], corresponding to the stretching vibration peak of Ce-O-Ce and the vibration of δ-C-H and Ce–O bonds, respectively. When the temperature rises to 600 °C, the absorption intensity of P=O (1197 cm^−1^) and C-O-C (1274 cm^−1^) are weakened. The rest of the P element and aromatic compounds remain in the condensed phase and play a role in the formation of char residue [35]. SEM was used to characterize the morphology of SPC, PA-SPC and 600 Ce-SPC.

As shown in Figure 2a,b, the SPC samples display a hollow rod-like structure. This structure is composed of many thin-layer structures, which resemble a honeycomb cluster structure. After being treated with phytic acid, the PA-SPC materials exhibit a similar porous parallel channel array structure (Figure 2c,d). This structure is profitable for the deposition of CeO_2_. However, the surfaces of PA-SPC are rougher compared to SPC. The EDS mapping images of the SPC and PA-SPC samples are shown in Figure 2e–h and Figure 2i–m, respectively. It can be noted that the SPC samples contain C, O and N elements. In the PA–SPC samples, the P element can be examined, due to the treatment of phytic acid and the existence of P–O in phytic acid, which is consistent with the FT–IR results. The CeO_2_/SPC is displayed in Figure 3a–f. It is clear that the rod-like structure of SPC is maintained, and the CeO_2_ materials are evenly deposited on SPC. Moreover, it is found that CeO_2_ is more uniformly distributed on the surface of SPC with an improvement of the calcined temperature. The EDS mapping image of 600 Ce-SPC proves the presence and even distribution of C, O, P, Ce and N elements (Figure 3g–l), indicating a uniform dispersion of CeO_2_ on the SPC surface.

The surface elements’ chemical characters of 600 Ce-SPC were unveiled by XPS. The full spectrum of 600 Ce-SPC is displayed in Figure 4a, and the presented peaks correspond to C, N, O and Ce. In Ce 3d (Figure 4b), the peaks are divided into four peaks, which are located at 898.9 eV, 901.7 eV, 883.5 eV and 917.3 eV [33]. The Ce3d are located at 898.9 eV and 901.7 eV [23] (Figure 4b), which is consistent with the presence of n-type Ce^3+/4+^O. The major intensity peaks for Ce 3d_5/2_ and Ce 3d_3/2_ are found at 883.5 eV and 898.9 eV, respectively [19]. The distance between Ce 3d_5/2_ and Ce 3d_3/2_ is 18.3 eV, which indicates that Ce^3+/4+^O is present in the n-p-n system [20]. Figure 4c reveals the C 1s states; the peaks at 284.8 eV, 285.7 eV, 286 eV and 293.23 eV are ascribed to the C-C, C-N, C-O and Plasmon loss features, respectively [36]. According to Figure 4d, the spectrum of N1s is split into four peaks at 398.9 eV, 394.3 eV, 400.7 eV and 404.8 eV. The peaks at 398.9 eV and 394.3 eV are concomitant with pyrrolium nitrogen cations and neutral nitrogen in the p-type PPy [37]. Small peaks at a higher BE between 405 and 407 eV are caused by oxidized nitrogen species [38]. The peak positioned at 400.7 eV is typically assigned to pyrrolic-N, which can thus be called by a generalized name: “hydrogenated nitrogen” [38]. The binding energy at 404.8 eV indicates the existence of interstitial N doping as well as the formation of N-O-Ti species [39]. In Figure 4e, the O 1s spectrum shows three oxygen contributions, denoted as O1 (528.2 eV), O2 (530.3 eV) and O3 (532.5 eV). The peaks at 528.2 eV, 530.3 eV and 532.5 eV are ascribed to metal–oxygen bonding, a large number of defect sites with low oxygen coordination, and a hydroxyl group on the surface-adsorbed water molecules, respectively [40].

The N_2_ adsorption–desorption measurement was applied to study the special surface area of 600 Ce-SPC. As shown in Figure 4f, according to the IUPAC classification, the isotherm of 600 Ce-SPC is a type III isotherm with type H3 hysteresis loops [39], indicating a mesoporous structure. Meanwhile, the BET-specific surface area is 89.1421 m^2^/g.

In order to understand the effect of the conductivity of the contact interface on charge transfer in the catalyst, the open–close cycle was carried out by the photocurrent response test, as well as the electrochemical impedance test in ring mode. As can be seen from Figure 5a, the photocurrent value of the 600 Ce-SPC catalyst is higher than that of the 500 Ce-SPC and 450 Ce-SPC catalysts, indicating that the 600 Ce-SPC catalyst has the best charge separation performance. This indicates that the presence of carbon in the contact interface improves electrical conductivity.

The migration efficiency of electrons has a significant impact on photocatalytic performance. The charge transfer resistance of the prepared samples was obtained using electrochemical impedance spectroscopy. It is well known that the smaller arcs, the faster the separation of carriers. According to Figure 5b, the 600 Ce-SPC samples display the smallest semicircle radius, implying that 600 Ce-SPC has a higher efficient charge separation efficiency. This may be due to the closer combination of CeO_2_ and SPC with the increase in calcination temperature, promoting the separation of the carriers. The results indicate that the 600 Ce-SPC samples have better photocatalytic performance.

Figure 5c shows that the zeta potential is about 50 when the pH is 7, which demonstrates that our material has excellent stability. At high potential, the material itself forms nano-bubbles, which have a large surface area and surface charge and can absorb and remove surface pollutants.

The fluorescence emission energy is released when the electrons and holes recombine. Smaller intensity reflects a lower recombination rate. The room temperature photoluminescence (PL) spectra of the samples are displayed in Figure 5d. The emission peaks of all the prepared samples are observed around 400 and 283 nm. The PL intensity of the materials increases successively: 600 Ce-SPC < 500 Ce-SPC < 450 Ce-SPC < PA-SPC and SPC. The SPC and PA-SPC have the biggest intensity, indicating an easy combination. After the deposition of CeO_2_, the intensity of the 600 Ce-SPC, 500 Ce-SPC and 450 Ce-SPC samples are all lower than PA-SPC and SPC, indicating that the disposition of CeO_2_ is a benefit to increasing the separation of electrons and holes. On the other hand, the 600 Ce-SPC samples have the lowest PL intensity, implying the lowest electron-hole recombination and highest photocatalytic performance. This result is consistent with the EIS result. Meanwhile, although it is difficult to judge the heterostructure of Ce-doped biochar, we carefully studied the related literature [41,42] and believe that the heterostructure of our catalyst can be judged by the heterostructure of cerium oxide. Previous studies in the literature have shown that, usually, the positions of band edges for CeO_2_ use Mott–Schottky plots and obtain UV-Vis DRS results. The Mott–Schottky plots of Ce showed positive slopes, and 600 Ce-SPC was an n-type semiconductor [43,44].

### 3.2. Photocatalytic Activity

The photocatalytic properties of the SPC, PA-SPC and 600 Ce-SPC samples were investigated by degradation TC under visible light and dark reaction for 60 min. As displayed in Figure 6a, there is no obvious difference in the adsorption capacity of PA-SPC and 600 Ce-SPC, which indicates that light irradiation is requisite for PA-SPC and 600 Ce-SPC for the photocatalytic degradation of TC. However, in the first 60 min, SPC showed a transient adsorption capacity, after which the degradation rate did not decrease, and showed no light response. Moreover, it can be observed that the degradation efficiency of TC is about 20% at 60 min for PA-SPC under light irradiation. It was found that the original cerous nitrate hexahydrate did not work well under photocatalytic conditions. After 60 min, the degradation rate is only 19%. Further, after combination with CeO_2_, the degradation efficiency of 600 Ce-SPC is enhanced, which is almost 99% of TC at 60 min under light irradiation.

The effect of different calcination temperatures on the catalysts was also studied, and the result is displayed in Figure 6b. It is worth noting that the degradation efficiencies of 600 Ce-SPC, 500 Ce- SPC and 450 Ce-SPC were 99.98%, 82.46% and 63.2%, respectively. It can be concluded that 600 Ce-SPC has the best performance. The FTIR analysis suggested that 600 Ce-SPC was endowed with more condensed aromatic carbon structures and more available polar functional groups. In general, more aromatic carbon structures and more polar functional groups are related to 600 Ce-SPC’s increased TC degradation capability in soil [45].

The pseudo-second-order models were applied to fit the experimental data, as displayed in Figure 6c. The degradation rate constant k of 600 Ce-SPC was k = 0.10095, which is 3.28 times than that of 500 Ce-SPC and 5.88 times of 450 Ce-SPC. These results indicate that the temperature of 600 °C has an obvious effect on the improvement of the photocatalytic efficiency of Ce-SPC. It may be that the formation of a heterojunction between the two inhibits the rapid recombination of the photogenerated carrier, and thus, improves the catalytic efficiency of the composite.

The effect of different catalyst dosages of the 600 Ce-SPC catalyst on the degradation of TC was investigated. According to Figure 6d, it can be said that the degradation efficiency of TC is 76%, 87% and 99% at 60 min for the 10 mg, 15 mg and 20 mg catalyst, respectively, indicating that the increased content of the catalyst is profitable for increasing degradation efficiency. The result implies that a higher catalyst dosage possesses a higher active substance, which leads to higher degradation efficiency.

The effect of different initial pH values on the degradation of TC by 600 Ce-SPC is shown in Figure 6e. It must be noted that the degradation efficiency improves when the pH value increases from 3 to 7. However, the degradation efficiency decreases when the pH value further increases to 9 and 11. The degradation efficiency of TC is 78%, 87.4%, 96%, 82% and 79.3% for pH values of 3, 5, 7, 9 and 11, respectively. The result displays that the optimum pH value is 7. This may be because rare earth metals, as metal oxides, belong to amphoteric compounds. Strong acid and base environments will have a certain impact on these systems, but the catalyst under different pH conditions still shows strong adaptability. Due to the existence of electrostatic repulsion between anion and 600 Ce-SPC in the reaction system with pH = 7, the contact opportunity of anion and 600 Ce-SPC is reduced. However, electrically neutral tetracycline is more likely to contact 600 Ce-SPC through electrostatic attraction and then be catalyzed by active components under visible light irradiation.

Natural organic substances (NOM) are always present in natural water bodies and have a significant impact on advanced oxidation processes. Therefore, the effect of different humic acid (HA) concentrations on the oxidative degradation of TC by the 600 Ce-SPC photocatalyst was studied, and the results are shown in Figure 7a. When the concentration of HA is 0.01 mg/L, 0.05 mg/L and 0.1 mg/L, the degradation efficiency of TC is 96.2%, 93% and 89% after 60 min, respectively. The results show that the addition of HA has an insignificant inhibitory effect on the reaction system, which can competitively scavenge free radicals in the NOM molecule by these reaction sites. As the Ce-SPC photocatalyst attacks and degrades TC by producing 1O_2_, almost no free radicals are produced. The effects of Cl^−^, (Figure 7b) NO^3−^ (Figure 7c) and (Figure 7d) on the oxidative degradation of TC by the Ce-SPC photocatalyst were also investigated, and the results are shown in Figure 7b–d. The degradation efficiency of TC in the 600 Ce-SPC, Cl^−^/600 Ce-SPC, NO_3_^−^/600 Ce-SPC and HCO_3_^−^/600 Ce-SPC system was about 95%, 94% and 93.2% at 60 min, respectively. The degradation efficiency of 600 Ce-SPC in an anion reaction system showed a downwards trend in comparison with that of 600 Ce-SPC without anions. After the photocatalytic reaction, the same trend appeared in the four experimental groups containing Cl^−^, NO^3−^ and HCO^3−^ in the reaction system containing Cl^−^, NO^3−^ and HCO^3−^.

Normally, hydroxyl radicals (·OH), sulfate radicals (SO^2−^), superoxide radicals (·O^2−^), photo-rising holes (h^+^) and photo-rising electrons (e^−^) are mainly reactive oxygen species (ROS) in the photodegradation of TC. In this study, methanol was used as a scavenger for hydroxyl radicals (·OH) and sulfate radicals (SO^2−^); tert-butanol (TBA) was used as a scavenger for hydroxyl radicals (·OH) and sulfate radicals (SO^2−^); p-benzoquinone (BQ) was used as a scavenger for superoxide radicals (·O^2−^); and furfuryl alcohol was used as a scavenger for reactive oxygen species (ROS), photo-rising holes (h^+^) and photo-rising electrons (e^−^), respectively. As shown in Figure 8a, the addition of BQ and TBA has a moderate inhibitory effect on TC degradation. The addition of methanol has a mild inhibitory effect on TC degradation, which proves that ·OH and SO^2−^ exist. However, these two active substances are not dominant in this system. Furfuryl alcohol can effectively capture e^−^ and further inhibit the TC removal rate in a light-Fenton reaction (decreasing by about 40%). Furfuryl alcohol significantly affected the TC removal rate (from 99% to 49.7% within 60 min). Since the scavenger furfuryl alcohol did not completely eliminate the degradation properties of TC, this implies that singlet oxygen/superoxide radical/hole transfer species play a major role in the oxidation of pollutants.

In order to evaluate the stability of the 600 Ce-SPC, four continuous repeated catalytic tests were conducted under light irradiation. As shown in Figure 8b, the results present a certain decline after four cycles. However, all the degradation efficiencies are more than 70%, indicating the stability of 600 Ce-SPC.

The proposed photodegradation mechanism of TC is represented in Figure 9 [21,46]. The main function of biochar is to capture the electrons in the conduction band (CB) of CeO_2_. There are several metals in the 600 Ce-SPC gallery with empty d orbitals that accept electrons as π-acceptor metals. These electron acceptors enhance the electron transfer rate at the CeO_2_ interface and inhibit the recombination of electron–hole pairs [47]. At present, Ce 600-SPC is a new type of semiconductor catalyst with low band gap energy (~2.3 ev), unique light absorption capacity and strong photooxidation capacity. SPC and CeO_2_ can form a type i heterojunction, which can effectively promote the separation of charge carriers, so as to broaden the utilization range of solar energy and improve the photocatalytic activity [48]. As a result, adsorbent O_2_ molecules accept the captured electrons and produce reactive oxidants, including H_2_O_2_ and O_2_^•−^. As a result, the life of the hole is enhanced. Narrow band gap energy and excellent visible light absorption ability improve the photoelectronic transition performance, and 600 Ce-SPC heterojunction is more conducive to the separation of h^+^,·O^2−^,·OH. Under visible light irradiation, e^-^ and h^+^ produced in the conduction and valence bands of 600 Ce-SPC will migrate to the conduction and valence bands of CeO_2_ and produce free radical catalytic oxidation of tetracycline pollutants. The excellent photocatalytic performance of 600 Ce-SPC mainly depends on the improvement of visible light absorption and the rapid e^−^/h^+^ separation efficiency. The degradation process of tetracycline and its possible by-products is displayed in Figure 9b [49].

## 4. Conclusions

In this study, a novel biological template for the synthesis of a ceria-modified soybean carbon Ce-SPC photocatalyst was developed, which has excellent catalytic performance under xenon lamp irradiation. The modified soybean powder carbon material was successfully loaded with cerium dioxide through the methods of freeze-drying and recalcination. The results show that the degradation efficiency of 10 mg/L of TC at 20 mg of 600 Ce-SPC was about 99.98% at 60 min under light irradiation, indicating that the 600 Ce-SPC catalyst has high photocatalytic activity. A low-cost, green and efficient photocatalytic degradation material was prepared. The work is expected to elucidate the mechanism of the photocatalyst enhancement of CeO_2_ modified carbides and provide a new feasible strategy for the synthesis and application of photocatalysts.

## Figures and Tables

**Figure 1 nanomaterials-13-01076-f001:**
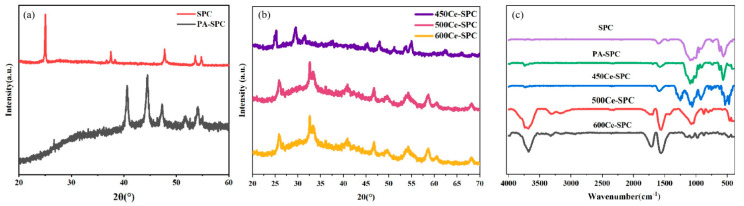
XRD patterns of pure SPC and PA-SPC composite catalysts (**a**), XRD patterns of 450 Ce–SPC/500 Ce–SPC and 600 Ce–SPC composite catalysts (**b**), FT–IR image of the different SPC samples (**c**).

**Figure 2 nanomaterials-13-01076-f002:**
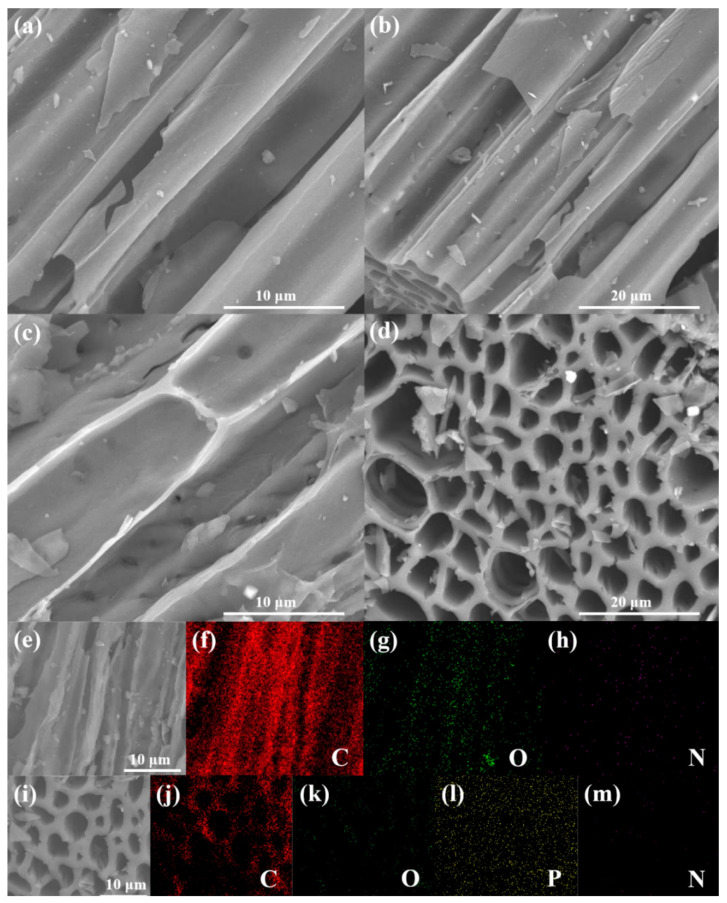
The SEM image of SPC (**a**,**b**), PA–SPC (**c**,**d**), EDS elemental mapping images of SPC (**e**–**h**) and PA–SPC (**i**–**m**).

**Figure 3 nanomaterials-13-01076-f003:**
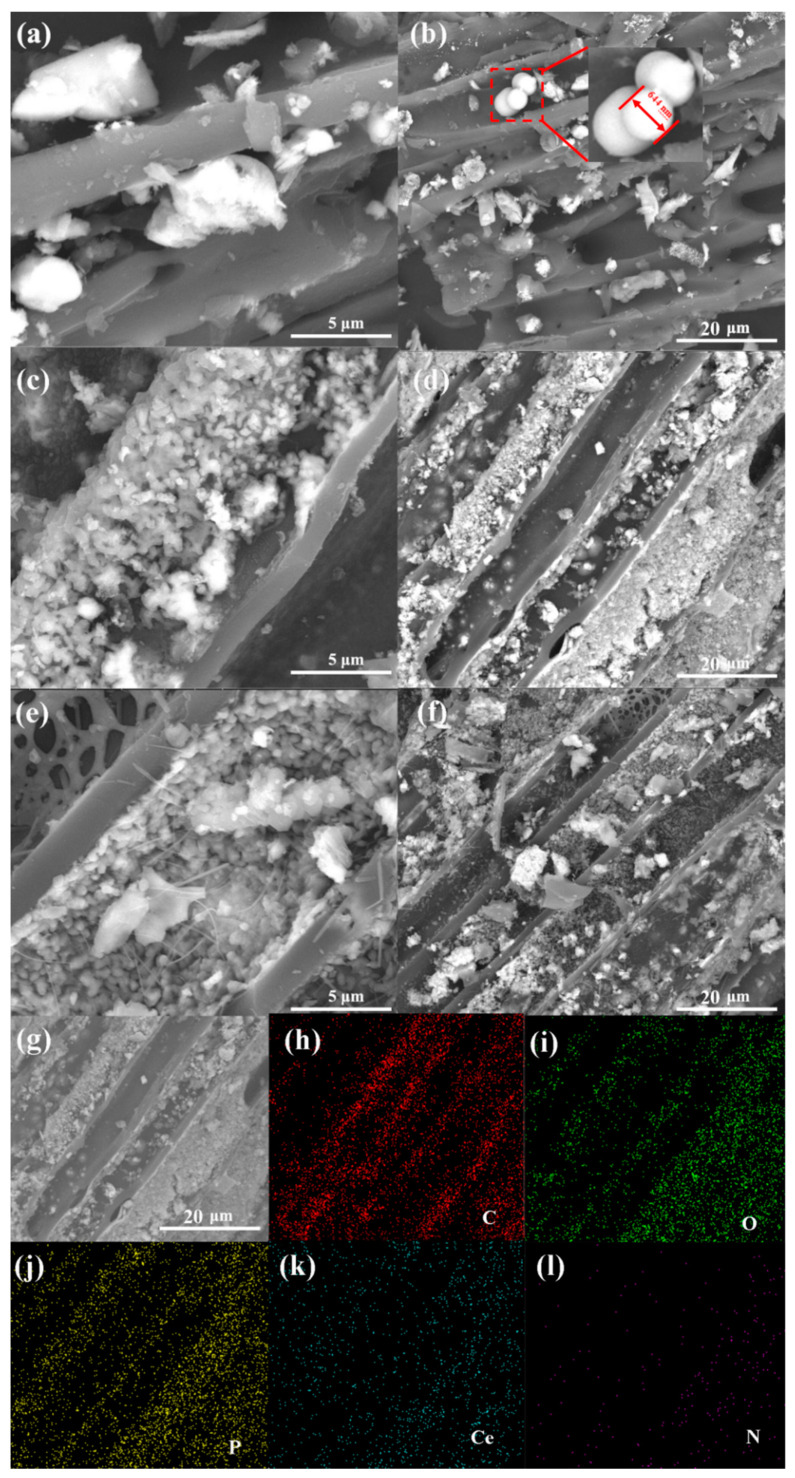
SEM (**a**,**b**): 450 Ce-SPC (**c**,**d**): 600 Ce-SPC (**e**,**f**): 500 Ce-SPC, EDS elemental mapping image of 600 Ce-SPC (**g**–**l**).

**Figure 4 nanomaterials-13-01076-f004:**
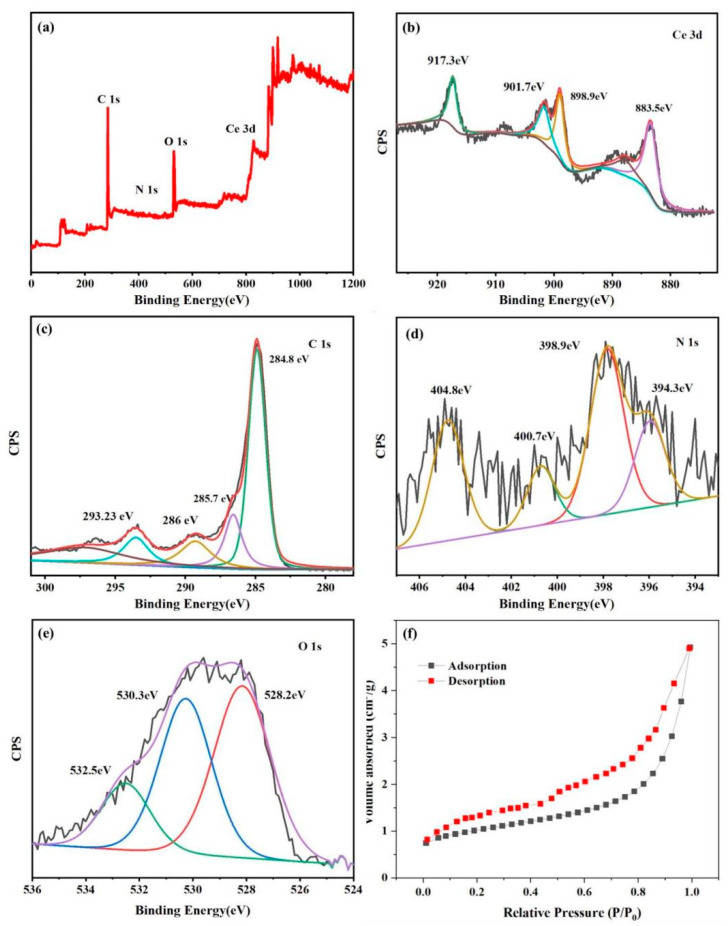
XPS spectra of CeSPC-600. (**a**) Survey spectrum; (**b**) Ce 3 d; (**c**) C 1 s; (**d**) N 1 s; (**e**) O1 s of 600 Ce-SPC; (**f**) N_2_ adsorption/desorption isotherm curves for of 600 Ce-SPC.

**Figure 5 nanomaterials-13-01076-f005:**
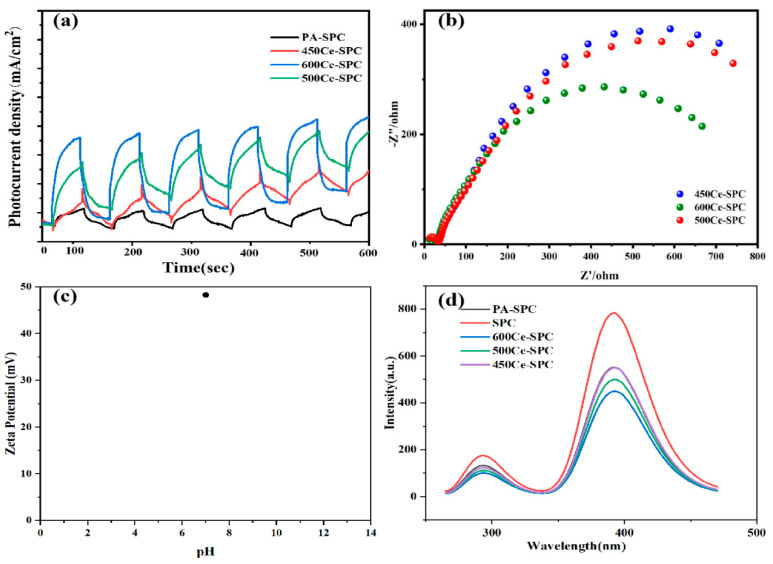
(**a**) Photocurrent responses; (**b**) EIS Nyquist plots; (**c**) zeta potential of 600 Ce-SPC; (**d**) photoluminescence spectra of SPC, PA-SPC, 600 Ce-SPC, 500 Ce-SPC and 450 Ce-SPC nanocomposites.

**Figure 6 nanomaterials-13-01076-f006:**
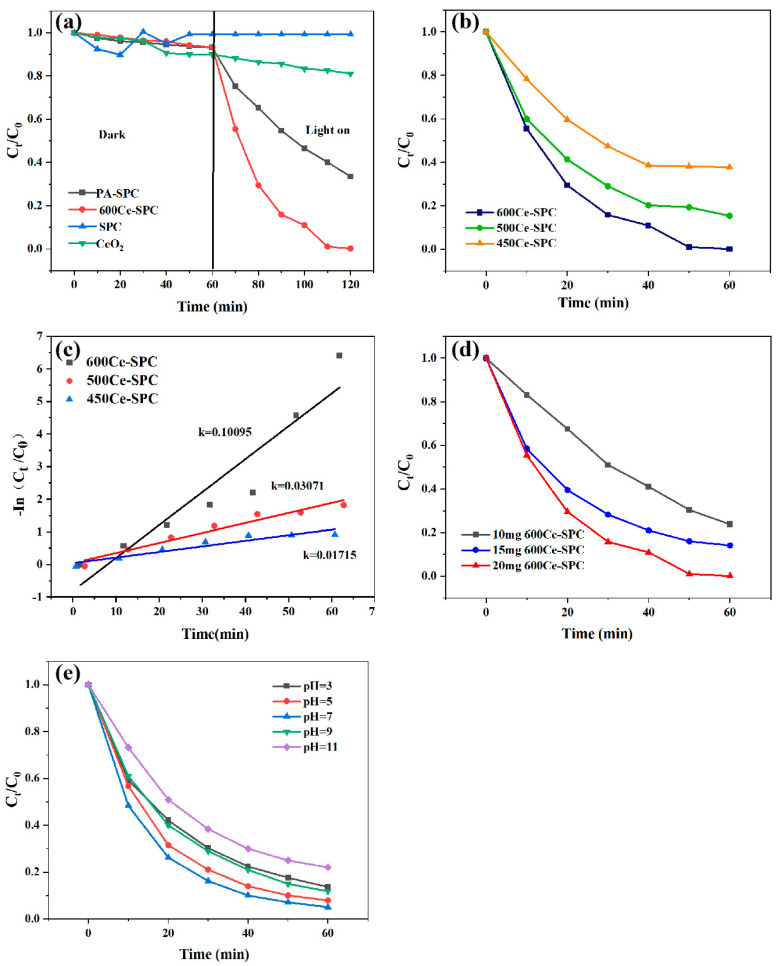
(**a**) Photocatalytic adsorption–degradation performance of SPC, PA–SPC, CeO_2_ and 600 Ce–SPC (**b**) Photocatalytic activity of the photodegradation of TC for 60 min by 450 Ce–SPC/500 Ce–SPC/600 Ce–SPC. (**c**) Secondary kinetics of photodegradation of TC for 60 min by 450 Ce–SPC/500 Ce–SPC/600 Ce–SPC. (**d**) Photocatalytic performance of different catalyst dosages of 600 Ce–SPC, (**e**) Effect of pH on photocatalytic experiment by 600 Ce–SPC.

**Figure 7 nanomaterials-13-01076-f007:**
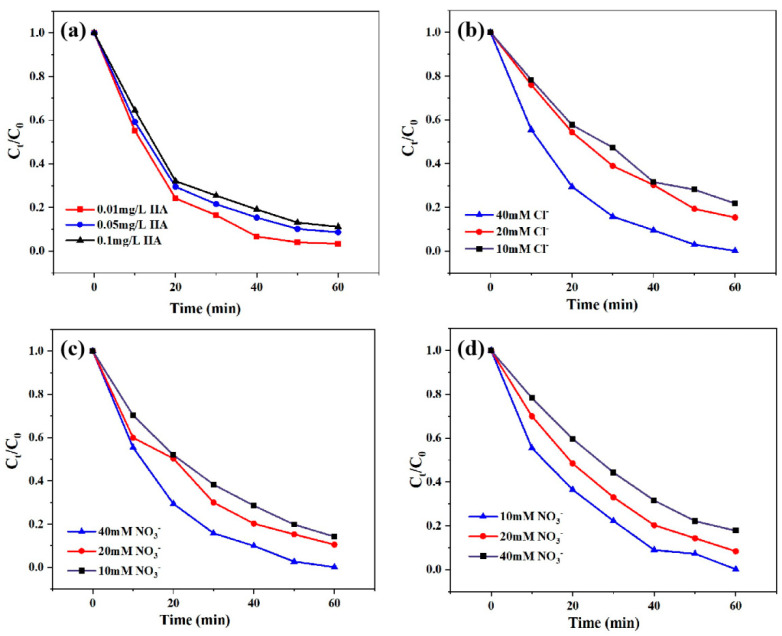
(**a**) Effect of humic acid on photocatalysis. (**b**) Cl-effect on photocatalysis. (**c**) NO^3−^ effect on photocatalysis. (**d**) HCO^3−^ effect on photocatalysis.

**Figure 8 nanomaterials-13-01076-f008:**
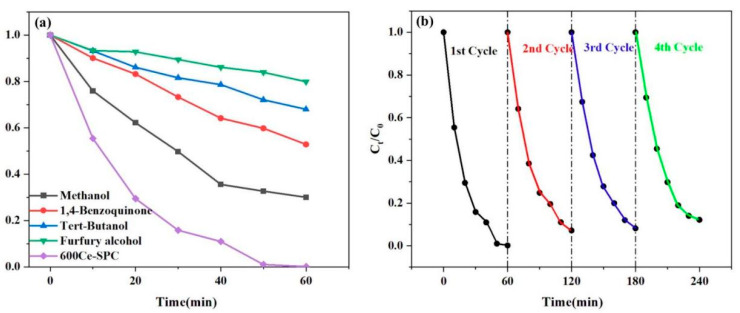
(**a**) The degradation of TC in the active species exploration of methanol with 600 Ce-SPC. (**b**) The reusability of the 600 Ce-SPC samples across four cycles for degradation TC.

**Figure 9 nanomaterials-13-01076-f009:**
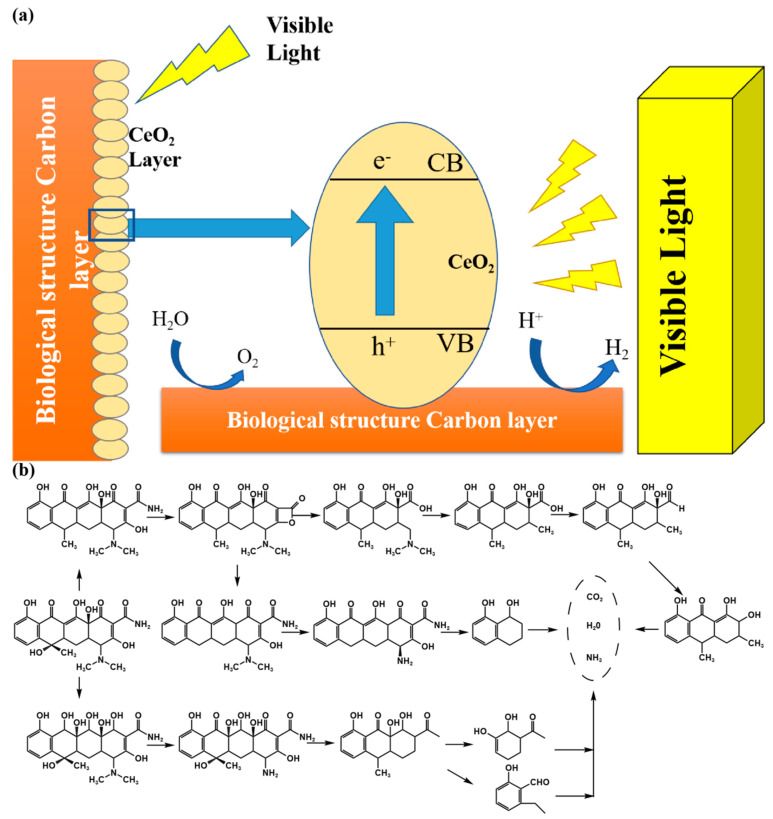
(**a**) 600 Ce-SPC diagram of photocatalytic mechanism. (**b**) Possible degradation pathways of TC in sample 600 Ce-SPC photocatalytic system.

## Data Availability

Data of the present study are available in the article.

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
