# Peer review of "Degradation of Tetracycline with Photocatalysis by CeO2-Loaded Soybean Powder Carbon"

_nanomaterials, 2023, doi:10.3390/nano13061076_

Round 1

Reviewer 1 Report

Manuscript of Yu and co-workers reports the preparation of CeO2 deposited on soybean powder carbon material modified with phytic acid for the photodegradation of tetracycline. Two important points, in my opinion, preclude the publication in Nanomaterials:

·         Some important and very recent reviews describe the use of nanomaterials for the tetracycline photodegradaion (Journal of Hazardous Materials (2022), 423(Part_B), 127145 , Chemosphere (2022), 287(Part_2), 132234 , Environmental Science and Pollution Research https://doi.org/10.1007/s11356-022-19766-y). Thus I believe that this work lacks of novelty required by Nanomaterials

·         Considering the dimensions of these object, I am not sure that "Nanomaterial" is the correct journal for this publication.

In addition:

·         Typos and grammatical errors must be corrected

·         Some references about tetracyclines and their use are required

·         From the introduction section, the novelty of this work is not clear.

·         Section 2.3 must be described

·         I do not understand the correlation between NOM and the HA concentrations

Reviewer 2 Report

In this paper, the authors investigated the photocatalytic activity of composites containing CeO2 and soybean powder carbon material. They demonstrated that the optimal photocatalyst can degrade 99% of tetracycline (TC) during 60 min illumination. I believe that the manuscript could be published in Nanomaterials, but major revisions are required:

1. I would recommend to throughly revise the manuscript on the language, since a huge number of spelling mistakes were detected in the whole paper.

2. Also, the formatting and quality of presentation are very poor which makes it difficult to read and evaluate this research.

3. The authors declare that they performed UV-Vis DRS analysis, but I did not find the results as well as the band gap value of the prepared materials.

4. It is necessary to investigate products formed during the photooxidation of TC because they may be more toxic than the initial antibiotic.

5. It would be great if the authors could evaluate the mean particle size of the photocatalysts (for instance, based on SEM).

6. The authors should investigate the photocatalytic activity of the pure CeO2 and compare the results with the composites.

7. A table, where a comparison of this study with other investigations, should be included. This could help to assess the promise of the developed photocatalyst.

8. This reviewer did not understand what was the point of including only a Sub-section with a description of the materials characterization in the Supplementary file. This information could be added in the main text.

9. What is the support role in the prepared composites for boosting the photocatalytic activity? A detailed desciption should be included in the manuscript.

10. Continuing the previous remark, the authors proposed that the heterojunction can be formed, preventing the rapid recombination of the photoinduced charges. In this regard, it is necessary to indicate what type of heterojunction (type-I, II, III, Z-scheme, or S-scheme) is formed (see please for consulting and citing: 10.1002/jctb.7091; 10.1002/adma.201601694).

Reviewer 3 Report

Comments to the author

Author has prepared Degradation of tetracycline with photo-catalysis by CeO2 loaded soybean powder carbon

The studies were well carried over, the figures are well structurally organized and arranged. I have few concerns and comments that need to be clarify/ justify before prior to publication.

Author has prepared Degradation of tetracycline by proposed catalyst, please describe the disadvantage of soybean powder carbon as alone and incorporate your proposed catalyst of CeO2 loaded soybean powder carbon photo-catalyst their advantages, so that this article reader can understand more specifically.

I would suggest to conduct zeta potential measurements at fixed pH, later you can explain that how your proposed catalyst material that can interact with organic pollutant of tetracycline.

Please author need to be more careful, during the advanced Oxidation Processes (AOPs) it’s generate hydroxyl radicals OH and also other reactive oxygen species, How do you confirm your proposed materials CeO2 loaded soybean powder carbon that was produced hydroxyl radicals OH?? Have you performed any supporting evidence are trapping theses radicals by electron paramagnetic resonance (EPR)??

Please provide the band gap details of your proposed materials of CeO2 loaded soybean powder carbon, and discuss about how this band gap are helping to your photo-catalytic studies?!!!.

For more information please read and refer the following research articles are especially band gap studies information.

Magnetically separable nano-spherical g-C3N4@Fe3O4 as a recyclable viable material for chromium adsorption and visible light driven catalytic reduction of aromatic nitro compounds, ACS Sustainable Chemistry & Engineering, 2019, 7, 6662-6671.

Please provide double beam UV-Visible spectrophotometer for before initiation of tetracycline, subsequently for each time interval how this spectrum were decreased in following concentration were decreased during the degradation as well as in different time of intervals (kinetics), so that this article reader can clearly understand of this studies and compare your kinetics data with your proposed material against previously published literature data’s.

I suggest author to read and refer the following research articles are especially comparison tables will be useful to the author information’s and also UV-Visible spectroscopic studies

Two in one: Poly(ethyleneimine)-modified MnO2 nanosheets for ultrasensitive detection and catalytic reduction of 2,4,6-Trinitrotoluene and other nitro aromatics, ACS Sustainable Chemistry & Engineering, 2021, 9, 1142-1151.

Heavy metal and organic dye removal via a hybrid porous hexagonal boron nitride-based magnetic aerogel, npj Clean Water, 2022, 5, 24.

Author must pay attention about the notation of superscript and subscript that need to fix it throughout the manuscript.

Round 2

Reviewer 1 Report

authors addressed all points raised during the first step.

Reviewer 2 Report

The manuscript has been carefully revised according to my recommendations. I would suggest the acceptance.

Reviewer 3 Report

The reviewer thanks to the authors for this revised version that address most of the comments made in its previous report. The manuscript can be published as it stands.